# Organic Rice Production Practices: Effects on Grain End-Use Quality, Healthfulness, and Safety

**DOI:** 10.3390/foods12010073

**Published:** 2022-12-23

**Authors:** Christine Bergman, Mhansi Pandhi

**Affiliations:** Food & Beverage and Event Management, University of Nevada Las Vegas, Las Vegas, NV 89154, USA

**Keywords:** rice, organic, sustainable, quality, safety, nutrition, pesticides, mycotoxins, gamma-oryzanol, tocols

## Abstract

Demand for rice labeled as organic is growing globally. Consumers state that foods labeled as organic are nutritionally superior and safer than their conventionally produced equivalent. The research question for this systematic review is as follows: is there a difference between the effects of sustainable agriculture and conventional farming methods on rice grain end-use quality, healthfulness, and safety? The studies (*n* = 23) examined for this review suggest that organic production practices don’t influence most end-use quality (e.g., chalk, milling yield, pasting properties) and healthfulness (e.g., gamma-oryzanol fraction and tocols) traits studied, or if it does, it will be to a small degree. If differences in end-use quality traits are found, they will be associated with grain protein content, which varies along with the dose of nitrogen applied during rice growth. We conclude that the studies evaluated in this review found that organically produced rice grain was less likely to contain residues of the pesticides (e.g., organochlorine) examined in the study than the rice is grown using conventional methods. There was some evidence that organically grown rice is more likely to be contaminated with mycotoxin-producing fungi and some mycotoxins. Common shortcomings of the studies were that they were poorly designed, with limited to no details of the cultural management practices used to grow the rice studied, the length of time fields was under organic management not stated, cultivars were not named, and the data wasn’t analyzed statistically.

## 1. Introduction

Rice is the staple food for more than half of the world’s population and is grown in over 100 countries [1]. The International Rice Information System indicates there are at least 5000 released rice varieties and many more if traditional varieties are considered [2]. Consumers generally choose a rice type with the cooking and sensory properties they are accustomed to or that suits the meal they will be preparing. For example, jasmine-types with their buttery popcorn-like aroma and soft texture are preferred by many people in Thailand and are served along with Thai-inspired meals prepared across the world. The cooking and sensory properties of rice and how it performs in processed products, such as canned soups, are known as rice end-use quality [3].

Most people across the globe eat rice in its milled (or polished) form. Rice that has been milled provides consumers with kilocalories, protein, vitamins, and minerals. Those that choose to eat unmilled (i.e., threshed or brown rice), commonly referred to as thrashed or brown rice, also obtain significant amounts of dietary fiber, lipids, and various phytonutrients [4]. An increasing number of consumers are choosing to eat unmilled rice and rice that has been grown using the principles of organic farming practices [5].

It has been proposed that there are two distinct schools of thought on how farming is practiced across the globe: the industrial and the agrarian philosophies [6]. Farmers and agronomy researchers tend to use the following categories to describe rice production methods: conventional production methods, sustainable agriculture techniques (e.g., organic, biodynamic, and regenerative), and the system of rice intensification.

The classifications of industrial or conventional farming “typically use synthetic pesticides, herbicides, and fertilizers, may use organic soil amendments; fields are frequently planted in short rotations” and generally uses monocropping systems [7]. Sustainable agriculture, as legally defined in U.S. Code Title 7, Section 3103 is an integrated system of plant and animal production techniques that have a site-specific application that will over the long term: Satisfy human food and fiber needs, enhance environmental quality and the natural resource base upon which the agricultural economy depends; make the most efficient use of nonrenewable resources and on-farm resources and integrate, where appropriate, natural biological cycles and controls, sustain the economic viability of farm operations and enhance the quality of life for farmers and society as a whole.

The UN Food and Agriculture Organization has defined organic agriculture as “a unique production management system which promotes and enhances agro-ecosystem health, including biodiversity, biological cycles, and soil biological activity, and this is accomplished by using on-farm agronomic, biological and mechanical methods in exclusion of all synthetic off-farm inputs” [8]. Thus, organic agriculture is a type of sustainable agriculture. Another form of sustainable agriculture is known as biodynamic farming. These types of farms generally grow several different crops, avoid the use of conventional inputs, produce and distribute the food in a decentralized manner, and take into consideration celestial and terrestrial influences on biological organisms [9]. A definition for regenerative farming has been proposed to be as follows: “an approach to farming that uses soil conservation as the entry point to regenerate and contribute to multiple provisioning, regulating and supporting services, with the objective that this will enhance not only the environmental, but also the social and economic dimensions of sustainable food production” [10].

Farmers that produce rice using sustainable production methods are often smallholders that perform low-input farming because it is their traditional way of farming, and they have limited resources to invest in conventional inputs [11]. Others, utilize sustainable methods to prevent the negative effects of conventional production methods that have been in use since the Green Revolution. Lastly, others have converted to using sustainable farming methods due to increased consumer demand for these foods and their willingness to pay premium prices for them [12]. These farmers in general adhere to their nation’s regulations on production practices required to allow foods to be labeled organic.

Between 2019 and 2025, the global organic rice market is expected to increase at a compound annual growth rate of 8% [5]. This increased demand is occurring globally, with the greatest increase in individual demand being in the European Union and North America. Consumers report that they purchase foods produced organically for the following reasons: healthier, helping the environment, and convenience [13]. For organic rice, the emotional route (e.g., I will feel happy if I buy organic rice) had a greater impact on its purchase intention than did the rational route (e.g., buying organic rice can form a good impression for me) [14]. Review articles report inconsistent evidence that foods produced using organic methods are significantly different in nutrient content compared to conventionally produced foods. However, organic foods are generally considered safer for consumption due to containing lower levels of pesticides and antibiotic residues [15].

The research question for this systematic review is as follows: is there a difference between the effects of sustainable agriculture and conventional farming methods on rice grain end-use quality, healthfulness, and safety?

## 2. Methods

This review was created according to the Preferred Reporting Items for Systematic Reviews and Meta-Analyses (PRISMA) Checklist and Guidance of the European Food Safety Authority [16]. This type of systematic review uses a clearly formulated question along with explicit steps to identify, select, and critically appraise previously published research and summarize the data from the studies found during the review.

### 2.1. Inclusion Criteria

The inclusion criteria for this study were experimental studies that evaluated the effects of sustainable farming production methods on rice grain end-use quality, healthfulness, and safety. Studies included those published during the previous 25 years (i.e., 1996–2021) and complete articles written in English. This time period was selected because it was in 2002 that the regulations under the U.S. Organic Foods Production Act were implemented and other countries such as Brazil adopted similar regulations sometime after this [15].

### 2.2. Information Sources

Five electronic databases that house different journals were searched; specifically, Academic Search Premier© (EBSCO Industries, Birmingham, AL, USA), Directory of Open Access Journals (Licensed under CC BY-SA 2.0), PubMed (US National Library of Medicine and National Institute of Health, https://pubmed.ncbi.nlm.nih.gov/, ScienceDirect ^®^ (Elsevier B.V., https://www.elsevier.com/solutions/sciencedirect, and Scopus^®^ (Elsevier B.V., https://www.elsevier.com/en-gb/solutions/scopus. All databases were accessed between 1 June 2022 and 5 June 2022.

### 2.3. Study Selection Process

Two authors followed the following protocol to identify articles for use in this study. The following keywords and Boolean operators were used to search the databases: #1. rice AND (#2. organic OR sustainable OR biodynamic or regenerative) AND (#3. nutrition OR health OR Vitamins OR minerals OR phytochemical OR quality OR cooking OR processing OR amylose OR safety OR Pesticide OR Herbicide OR Fungicide). Each search included one term from categories #1, #2, and #3.

The titles and abstracts of all articles identified from the selection process were screened for each one’s relevancy to the research question by two authors separately. They were independently screened, coded, and evaluated for suitability to be included in the study. Specifically, a study was selected if it evaluated the effects of growing rice using any sustainable agriculture methods on any aspect of rice grain end-use quality, healthfulness, safety, and methodological quality. Full articles for each study selected in the previous step were then evaluated using the exclusion criteria. The authors discussed any differences in the articles identified for inclusion in the study and came to an agreement on which to include and which to not include based on the exclusion criteria.

### 2.4. Exclusion Criteria

The following exclusion criteria were used: (1) full paper not written in English, (2) not specifically related to the research question, (3) no conventional rice used for comparison to sustainably produced rice, (4) non-research articles, (5) duplicates, and (6) methodology very unclear.

### 2.5. Data Analysis

A meta-analysis was not conducted for this review as the studies didn’t use similar experimental designs or analytical methods. The articles reviewed either didn’t describe the cultural management practices or the ones used were very different from the others reported. Therefore, only a qualitative analysis of the data collected from the studies was conducted, as recommended by the Cochrane handbook for systematic reviews of intervention studies [17].

Research articles included in the study were evaluated to determine if suitable statistical methods were used to analyze the data collected during the study.

### 2.6. Risk of Bias

Bias in study design was decreased by using five databases to identify the studies for the systematic review. However, grey literature such as conference proceedings was not included. Selection bias was limited by the creation and use of inclusion and exclusion criteria. Two authors, separately, performed the literature identification and selection process to limit personal bias. The risk of assessment bias was reduced by critically evaluating all of the studies included in the systematic review and discussing them in the discussion section below.

## 3. Results

### 3.1. Identification of Included Studies

The study selection process yielded 12,757 records (Figure 1). After the deletion of duplicates, 1028 records remained. The titles and abstracts of these records were evaluated to determine if they contained descriptions of experimental studies that evaluated the effects of any sustainable farming production methods on rice grain end-use quality, healthfulness, or safety. In addition, the dates were evaluated to determine if the records were published in the previous 25 years and were written in English. Author one and two, using these inclusion criteria identified 60 and 49 articles, respectively. The additional articles chosen by author two were found by author one to be studies related to the nutrient content of organic soils used to grow rice, not in rice grain. Therefore, 49 full articles were selected for further evaluation since soil nutrient content wasn’t one of the dependent variables being evaluated in this study. These 49 articles were then evaluated with the exclusion criteria. Of the 49 articles, 26 were deleted from the group due to one of the following exclusion reasons: full article not written in English, methods not specifically related to the research, no conventional agriculture farming dependent variable included, article duplication, or unclear methodology (Table 1). The primary reason for exclusion from the study was due to articles not being specifically related to the research topic (e.g., frog and rice co-culture using organic production methods). Thus, there were 23 articles found and deemed to be suitable for examination in this review study.

### 3.2. Study Characteristics

The studies included in this review were performed in the following countries: Afghanistan, Brazil, India, Italy, Malaysia, Philippines, South Korea, Spain & Portugal, Thailand, and United States (Table 2). The top countries involved in sustainable rice grain research were found to be Brazil and Thailand. Of the sustainable rice, grain-focused studies eight evaluated effects on end-use quality, nine on healthfulness, and 14 examined safety issues. All of the studies focused specifically on rice described as being organic by the authors. No other terminology used to describe sustainable farming practices was found in the articles.

Only Chen and McClung [18], Champagne et al. [19], Cho et al. [20], Tuano et al. [21], and VanQuyen et al. [22], reported in some detail the agronomic practices used to produce the rice for their study. Of these Chen and McClung [18], Champagne et al. [19], Tuano et al. [21], and VanQuyen et al. [22] also included the name of the rice cultivar(s) they studied and performed replications of their field treatments. Alves et al. [23] and Keawpeng et al. [24] reported the cultivar they studied, but limited information was included about the field management methods used. The other studies examined rice or rice products obtained from grocery stores and rice cooperatives.

Of the studies that evaluated end-use quality, four evaluated amylose content (Table 3) (Champagne et al. [19], Kaker et al. [25], Keawpeng et al. [24], Tuano et al. [21]), three examined protein content (Alves et al. [23], Kakar et al. [25], Keawpeng et al. [24]), and three examined lipid content (Alves et al. [23], Kakar et al. [25], Keawpeng et al. [24]. One study examined the sensory quality and volatile compounds (Champagne et al. [19]. Two studies determined the yield of head rice or whole kernels after milling (Alves et al. [23], Kakar et al. [25]) and one examined the percentage of stained grain (Kakar et al. [25]). Cooking time was measured by one study (Alves et al. [23]), as were elongation ratio, hardness, color, water uptake capacity, and starch crystallinity (Keawpeng et al. [24]). Kernel swelling power, H_2_O solubility, and starch X-ray diffraction patterns were each examined by one study (Keawpeng et al. [24]). Pasting properties were evaluated by two studies (Champagne et al. [19], Keawpeng et al. [24]). Rice noodle color, tensile strength, elasticity, cooking loss, rehydration ratio, aerobic plate content, and total yeast and mold contents were discussed in one paper (Thomas et al. [26]). One study evaluated kernel length, kernel breadth, and the ratio between the two before and after cooking (VanQuyen et al. [22]).

Studies focused on healthfulness traits included three that evaluated macro- and micro-mineral (i.e., element) content (Barbosa et al. [27], Champagne et al. [19], Poletti et al., 2014). Total phenolics were examined in three of the studies (Chen and McClung [18], Alves et al. [21], Tuano et al. [21]). One study examined flavonoid content (Chen and McClung [18]), and serum cholesterol, triglycerides, HDL-C, and LDL-C contents in rats (Mesomya [35]). Tocols and gamma-oryzanol were evaluated in two studies (Cho et al. [21], Chen and McClung [18], Tuano et al. [21]). The protein efficiency ratio, measured in rats, was also examined in one study (Mesomya et al. [36]).

Safety-related studies include six that evaluated total As levels (Barbosa et al., [27], Batista et al., [28], Hernandez-Martinez et al. [33], Juskelis et al. [38], Poletti et al. [40], Segura et al. [40]). Two studies evaluated inorganic As (Juskelis et al. [38]), Segura et al. [40]) and Cr levels (Barbosa et al. [27], Poletti et al. [33]). Hg levels were examined in one study (Hernandez-Martinez et al. [31]). Pb (Poletti et al. [38]) and Cd (Barbosa et al. [27]) were examined in one study each. Infestation by fungal pathogens was evaluated only by Alves et al. [23]. Various mycotoxins were examined in five studies (Alves et al. [23], Cirillo et al. [29], Gonzales et al. [30], Juan et al. [32], Ok et al. [37]). Two studies evaluated pesticide residues (Mesomya et al. [36], Rekha [39]).

### 3.3. End-Use Quality Traits

Alves et al. studied one long-grain cultivar with high amylose content [23]. The sample produced using conventional farming methods had a greater (*p* < 0.05) head rice yield (i.e., milling yield) than the grain harvested from a field managed using an organic cropping system. Another study examined perfect grains, which, according to their definition, was equivalent to head rice yield [25]. The cultivar examined had a greater (*p* < 0.05) head rice yield when grown using animal manure (organic treatment) or 50% animal manure + 50% sawdust fertilizer compared to the other treatments (i.e., conventional, sawdust, and 50% sawdust + conventional fertilizer treatments).

Champagne et al. studied cultivars that varied in amylose content from a low of 0% to a high of 21% [19]. The study found no difference in milled rice amylose content between organic and conventionally grown rice. The conventional management practices in terms of pesticides and herbicides weren’t described, however, the details of the organic practices were fully provided. Similarly, Tuano et al. found no effect of organic management on milled grain amylose content compared to conventional methods [21]. In this study amylose content of one of the cultivars examined across all treatments was 24.2% and the other was 13.8%. The study by Kakar et al. found amylose content (23%) to be higher in milled rice grown using animal manure plus 50% of the conventional treatment compared to all other treatments [25]. The lowest amyose content was reported in the rice grown using conventional management methods (20.9%). All of these studies evaluated samples collected from field experiments designed by the authors.

The protein content of milled rice grown using conventional methods was significantly greater (*p* < 0.05) than the rice grown using either 50% of the N in the conventional treatment or using organic methods [19]. Similarly, Keawpeng et al. found that conventionally produced rice had greater protein content compared to that produced using an organic method, 7.02 and 5.64%, respectively [24]. Alves et al. studied unmilled rice grown under organic methods and found that it was lower in protein (6.7%, *p* < 0.05) in comparison to its conventionally grown counterpart (7.8%) [23]. The protein content of unmilled rice was also examined by Kakar et al. [25]. They found that protein content was greater (*p* < 0.05) in grains produced using animal manure and 50% of the local recommended amount of N and P (8.75) and the organic treatment with 100% animal manure (8.0%) compared to the treatment using 100% of the recommended dose of N and P for conventional farming (7.6%). The study by Tuano et al. examined two rice crops, one in the wet season and one in the dry, the organically produced milled rice had less protein (*p* < 0.05, 6.3%) than those grown using conventional methods (8.8%) [21]. These studies all evaluated samples collected from field experiments designed by the authors.

Champagne et al. evaluated milled rice grown using three treatments: 100% conventional methods, 50% conventional N, and organic methods [19]. No significant difference (*p* > 0.05) was found for pasting viscosity properties between these treatments, except for one cultivar out of the five studied. That cultivar, Cypress, had a higher peak (*p* < 0.05) viscosity in rice grown organically (249 and 159 RVU, respectively) and using 50% of the conventional farming N level (244 and 156 RVU, respectively) in comparison to that grown using conventional methods (207 and 126, RVU, respectively). Keawpeng and Meenune reported that pasting peak temperature wasn’t different (*p* > 0.05) between organic and conventionally grown milled rice [24]. However, peak viscosity and setback were lower in the organic (117 and 119 RVU, respectively) than in conventional rice (124 and 130 RVU, respectively). Both of these studies evaluated rice that was by the authors using field production trials.

The sensory properties of milled samples of five cultivars (i.e., Cypress, Bengal, Jasmin 85, Jacinto, and Neches) were examined using descriptive sensory analysis, which included the assessment of 12 flavors and 14 textural attributes [19]. These cultivars were different cooking quality types and were grown using the following field management treatments: 100% nitrogen/conventional, 50% nitrogen/conventional, and organic. No differences were found in aroma due to production methods. Of the textural properties, slickness, hardness, and chewiness were slightly different (*p* < 0.05) for some of the cultivars grown using 100% nitrogen/conventional or 50% nitrogen/conventional versus organic production methods. Another study found no significant difference (*p* < 0.05) in the instrumental hardness of unmilled rice from the cultivar Phatthalung Sungyod grown using organic versus conventional production methods [34]. The rice examined in these two studies was grown by the authors using field production trials.

The cooking time of unmilled samples of cultivar IRGA 410 reportedly was 29.0 and 26.0 min for those grown using conventional and organic growing conditions, respectively [23]. These values weren’t analyzed statistically. No other studies evaluated the effects of field production methods on rice cooking time.

The kernel length of the cultivar Pusa Basmati 1 wasn’t different under seven different organic treatments in a randomized block trial with three replications. However, the kernel length-to-kernel breadth ratio in the organic treatments was greater than in the conventional treatments. The data wasn’t analyzed statistically, and the paper didn’t mention whether unmilled or milled samples were evaluated [22].

### 3.4. Healthfulness Traits

This study reviewed articles to determine if the authors found differences in Ca, Fe, and Zn levels in rice grown organically versus conventionally. One common finding was that the studies that identified the cultivars being studied identified a wide variation in mineral content between cultivars.

Champagne et al. found no significant (*p* < 0.05) difference in the level of Ca between milled samples of cultivars grown using organic and two conventional treatments (described above) [19]. Similarly, no significant (*p* < 0.05) effect of year was found on the Ca content of these samples. Effects of cultivar on Ca levels weren’t reported. In another study, Ca levels were significantly (*p* < 0.05) higher in all organic samples (*n* = 17; 103 mg kg^−1^) compared to those milled and produced conventionally (*n* = 33; 39.5 mg kg^−1^) [27]. These samples were all purchased from Brazilian grocery stores. No mention was made of whether the samples were unmilled or milled. Poletti et al. also studied samples purchased in Brazil [38]. They reported no significant difference in Ca levels in organic (*n* = 5) versus conventional (*n* = 9) samples.

Champagne et al. studied the Fe content of milled samples of several cultivars (described above) grown using organic and two conventional treatments (as described above) [19]. One cultivar, Jacinto, had significantly (*p* < 0.05) more Fe when grown under organic conditions (23 ppm) compared to the conventional one (15 ppm). No difference was found between the treatments for the other four cultivars. Similarly, Barbosa et al. reported no difference in Fe content between conventional and organically grown rice samples (as described above) [27]. However, in this case, the data wasn’t analyzed statistically. It should also be noted that these authors indicated the samples they evaluated had not been industrially enriched with Fe, which is common in some countries [41].

The studies that evaluated Zn all reported that the content of this mineral varied within organically grown rice and also within conventionally produced rice. Out of four cultivars examined by Champagne et al., only one was reported to have a significant difference in the amount of Zn between them when grown using organic and conventional cultural management practices [19]. The cultivar Jacinto had 22 ppm when grown using organic management practices while when produced using conventional methods it had 27 ppm. Another study of organic and conventionally produce rice found the former to have (16.9 mg kg^−1^) and the latter (21.1 mg kg^−1^), but the data wasn’t analyzed statistically [27]. Poletti et al. concluded that there were no differences in the Zn content of organic and conventionally grown rice, both unmilled and milled samples [38]. The data in this study weren’t analyzed statistically.

Alves et al. reported that organically grown unmilled rice had 33% more (*p* < 0.05) free phenolics (48 mg GAE equiv/100 g) than were found in the same cultivar grown under conventional field practices (32 mg GAE equiv/100 g) [23]. Unmilled rice (six cultivars) with various bran colors was obtained from organic and conventionally managed field trials grown over two years [18]. This study reported no difference (*p* > 0.05) in the level of total phenolics between the different field management methods, except for a cultivar (i.e., IL 121-1-1) with red bran during one year. The organic sample contained 5.86 mg GAE equiv/g and the conventionally grown one had 6.76 mg GAE equiv/g. These authors also studied flavonoid levels in the same samples and found similar results. Specifically, the only significant difference (*p* < 0.05) in flavanoid content was also found for IL 121-1-1. When grown organically (0.6 mg +-catechin equiv/g) it had lower flavanoid content compared to when grown conventionally (0.79 mg +-catechin equiv/g). In a study by Tuano et al. (2011), an organic fertilizer treatment was reported to be associated with lower total phenolics (*p* < 0.05) (30.7 mg g^−1^) for one cultivar in comparison to when it was grown using conventional field methods (37.7 mg g^−1^) [21].

Mesomya et al. evaluated the protein efficiency ratio of unmilled rice grown under organic and conventional field management using rats (10 rats per treatment plus a control). models [35]. No significant difference (*p* < 0.05) in the growth of the animals or protein efficiency ratio was found. Rats were also used to examine serum cholesterol, triglycerides, HDL-C, and LDL-C contents [35]. These authors using the same experimental design as in the previous study evaluated the effects of organic and conventional unmilled rice on the following rice serum lipid levels: serum cholesterol, triglycerides, HDL-C, and LDL-C. No significant difference (*p* > 0.05) in these lipid levels was found.

Gamma-oryzanol levels were found to be significantly higher (*p* < 0.05) in unmilled rice samples of cultivar Dongjin that had been grown using organic conditions (65.6 mg/100 g) compared to the cultivar grown using conventional management methods (60.2 mg/100 g) [20]. Both fields were in the same general location, therefore the rice matured under similar weather conditions. The organic field had been under South Korean organic management standards for five years. Gamma-oryzanol content was also evaluated in unmilled samples of three U.S. cultivars (i.e., Cocodrie, Presidio, and Sierra), a U.S. breeding line (IL 121-1-1), a giant embryo mutant, and one Indonesian cultivar (i.e., Sigoendaba) [18]. They were grown over two years, in both organically and conventionally managed fields. Cultivar and year effects had a significantly greater effect (*p* < 0.05) on gamma-oryzanol levels than the field management methods. Of these three effects, the cultivar had the greatest impact on the levels of this trait. Gamma-oryzanol levels tended to be lower in the genotypes grown using organic compared to conventional field management. But, this wasn’t true in all samples. For example, in one-year Sigoendaba had greater gamma-oryzanol levels in the organic versus conventionally grown samples. ‘NSIC Rc146′ and ‘NSIC Rc 160′ were grown over two years in the Philippines [21]. Organic fertilizer compared to conventional (e.g., inorganic) fertilizer and pesticide application versus none applied didn’t have a consistent effect on the gamma-oryzanol content of the samples. Conventional NSIC Rc146 grown with pesticide application had significantly (*p* < 0.05) more gamma-oryzanol compared to the organic samples. NSIC Rc160 grown using organic and conventional fertilizer with and without pesticide had similar levels (*p* < 0.05) of gamma-oryzanol. Replicate plots contributed significantly (*p* < 0.05) to the variation in sample gamma-oryzanol levels.

Cultivar and year effects had a significantly greater effect (*p* < 0.05) on tocol levels than did the field management methods for the samples studied by Chen and McClung and discussed above [18]. Cultivar had the greatest effect on the levels of tocols compared to the year the samples were grown in and the type of cultural management practices. No consistent effect was found on the levels of tocols in the unmilled samples described above in the study by Tuano et al. [21]. No effect of fertilizer type or pesticide application or not was found for the levels of tocols in the NSIC Rc146 samples. However, NSIC Rc160 grown using organic (157 mg 100 g^−1^ wet basis) versus conventional (76 mg 100 g^−1^ wet basis) fertilizer with pesticides had significantly (*p* < 0.05) higher levels of tocols. In addition, replicate plots contributed significantly (*p* < 0.05) to the variation in sample tocol levels.

### 3.5. Safety Traits

Barbosa et al. reported that they found higher levels of total As in conventional rice grain (*n* = 33, median 0.208 mg kg^−1^) in comparison to certified organic rice grain (*n* = 17, 0.158 mg kg^−1^) that was purchased from grocery stores [27]. The organic rice was certified by Brazilian IBD-Agricultural and Food Inspections. The data in this study weren’t analyzed statistically. Another study evaluating total As content found greater levels in organic milled rice (222.9 and 161.6 ng g-1, respectively) than in conventional milled rice [28]. In the same study, the mean of total As for the parboiled white samples was reported to be greater than the parboiled organic sample (214.9 and 174.1 ng g^−1^, respectively). Total As was reported to be higher in organic rice infant cereals compared to conventional infant rice cereal (154.9 and 96.3 μg/kg, respectively) [31]. The data in this study weren’t analyzed statistically. Another study that evaluated milled rice infant cereals reported indicated there was no difference between organic and conventionally produced cereals. The data in this study weren’t analyzed statistically [33]. Poletti et al. [38] and Segura et al. [40] evaluated organic and conventional milled rice samples and found no difference in total As levels. In addition, these authors reported no difference in total As levels between organic and conventional unmilled rice samples. All of the studies that evaluated total As used samples collected from stores, and none of them analyzed the reported data using statistical methods.

The levels of inorganic As in infant rice cereals that were obtained from grocery stores reportedly had no difference between those made using milled rice grown under organic and conventional production techniques [33]. Inorganic As was reported by Segura et al. to be 45% greater in organic milled rice compared to conventional milled rice and 41% greater in organic husked versus conventional husked rice [40]. The data in both of these studies weren’t analyzed statistically.

Cd was reported to be higher in conventionally (0.012 mg kg^−1^) grown rice samples purchased in grocery stores compared to ones produced using organic (0.005 mg kg^−1^) production methods [27]. However, this data wasn’t analyzed statistically.

Milled rice Cr levels were reportedly greater in conventional (3.0 mg kg^−1^) rice in comparison to that grown under organic conventional (2.3 mg kg^−1^) production methods [27]. On the contrary, Poletti et al. reported no difference in the level of Cr in organic versus conventional unmilled and milled samples purchased at grocery stores [38]. The data in both of these studies weren’t analyzed using statistical methodology.

Hg was higher in organic infant rice cereal compared to cereals made from conventionally grown rice (4.54 and 4.39 μg/kg, respectively) [31]. The Pb levels found in conventional and organic rice were similar [27]. On the contrary Poletti et al. reported that conventionally grown milled rice contained more PB (130 μg kg^−1^) than organically grown milled rice (98 μg kg^−1^). Barbosa et al. reported that the Cd content of conventionally produced conventional rice (0.012 mg kg^−1^) was higher than that grown under organic conditions (0.005 mg kg^−1^) [27]. All samples used in the studies described above were obtained from grocery stores and the data reported weren’t analyzed using statistical methods.

Alves et al. studied fungal pathogen levels in rough rice during storage [23]. They found that *Penicillium* sp., in organic rice was 4, 9, and 4 times higher in rough rice stored for 0, 6th, and 12th months, respectively, compared to rice produced under conventional systems. *Aspergillius* sp. was slightly higher in organic rough rice compared to organic prior to storage, while after six months of storage, conventional rough rice contained 70% more. Before storage, *Bipolaris* sp. was found in organic rice but not in the rough rice grown conventionally. The rice used for these studies was collected from field trials.

Deoxynivalenol, and fumonisin B1 and B2 were evaluated in various rice-based foods such as biscuits and breakfast cereals [29]. Deoxynivalenol and fumonisin B1 were reported to be higher in products made using conventionally grown rice (207 μg kg^−1^ and 205 μg kg^−1^, respectively) compared to that produces using organic rice (65 μg kg^−1^ and 150 μg kg^−1^, respectively). On the contrary, rice-based food products made using organically grown rice contained more fumonisin B1 (145 μg kg^−1^) than those produced using conventionally grown rice (30 μg kg^−1^). None of the data in this study were examined statistically.

Gonzales et al. studied the levels of ochratoxin A in milled rice and rice-based food products [30]. More organic samples contained ochratoxin A (30.0%, range 1.0 to 7.1 Ag/kg) compared to those produced using conventional growing methods (7.8%, range 4.3 to 27.3 Ag/kg). Juan et al. also evaluated the levels of ochratoxin A in conventionally and organically grown rice [32]. They found that some organically produced rice (4/9 samples, mean 2.57 ng/g) contained ochratoxin A, but none of the conventionally produced samples did (0/4 samples). No difference between organic unmilled rice and milled rice was reported. The data in this study weren’t examined statistically.

The occurrence of mycotoxins, specifically five 8-ketotrichothecene compounds, was studied in organically and conventionally produced rice [37]. They found no significant difference in the levels of deoxynivalenol, 3-acetyldeoxynivalenol, 15-acetyldeoxynivalenol, and fusarenone-X in conventionally and organically produced unmilled and milled rice. However, the levels of nivalenol were significantly higher (*p* < 0.05) in both organically produced unmilled and milled rice compared to the corresponding conventionally grown rice. The sample studies in this study were all obtained from grocery stores in South Korea.

Pesticide residues (i.e., p-nitrophenol, carbofuran, methyl parathion, and β-cyfluthrin) were measured in cooked rice samples of unmilled conventionally and organically produced rice that was obtained in Thailand [35]. The cooked rice samples contained p-nitrophenol (8.23 and 10.13 mg/kg, respectively). None of the other residues were found in the cooked rice sample. Also, none of the pesticide residues were identified in the serum from rats fed the cooked rice.

Rice obtained from organic and conventional farms in 16 regions of India was examined for pesticide residues [39]. The resides were from four groups of pesticides (i.e., organochlorine, carbamates, organophosphorous, and pyrethrites). The sites were chosen to represent all of India’s rice-growing regions in the northern and central parts of the country. Residues of organochlorines were present in all the conventionally grown rice samples. Organochlorine pesticide residues were found in two out of ten organic farms. These farms had both been converted from conventional to organic practices a few years ago. The presence of carbamates and pyrithroid were found in conventional rice samples, while no trace of either of these was found in the organic rice samples. Nothing related to the organophosphorous levels in rice was discussed in the paper.

## 4. Discussion

Rice grain characteristics result from differences in the genetics of the rice variety and environmental effects [3]. These effects include such things as the influence of climate, soil quality, seeding rates, field in-puts, grain processing, and grain storage. Thus, a well-designed study evaluating the effects of rice production practices would keep all of the cultivars the same. The inputs’ type, amount, and application time (e.g., soil amendments, soil type, fertilizers, pesticides, herbicides, and irrigation) would need to be reported in detail. Also, there must be cultivars and environmental replications.

This systematic review found only a relatively small number of studies examining sustainable agriculture practices’ effects on rice grain end-use quality, healthfulness, and safety. They all examined organic practices specifically. The rice cultivars used varied from study to study. In some studies, the cultivars were named; in others, they weren’t, as the rice samples were obtained from retail markets. Most of the studies lacked details on how the rice samples were grown, stored, milled, and packaged. Therefore, the results in this review should be taken with caution, as most of the studies have error rates from poor experimental design, limited or no reporting of how the rice kernels were processed and stored before analysis, or inadequate statistical analysis [42].

### 4.1. End-Use Quality

The translucence, shape, and uniformity are important aspects of rice end-use quality for consumers, millers, wholesalers, and retailers [43]. Surprisingly few studies in this review reported measuring aspects of grain appearance. Those that did examine rice grain appearance either found very small differences or none at all. Differences in grain color and chalk (i.e., opaque spots) between rice samples can often be seen using the naked eye. Therefore, had large differences in appearance occurred in the studies where these traits weren’t measured using instrumentation they would have still likely been reported. No such differences were mentioned in the studies reviewed for this paper. Chalk is known to have a genetic link and also an environmental cause, temperatures during grain filling increase its levels [44,45]. Future studies need to examine cultivars that are susceptible to developing chalk, expose them to low and high temperatures during grain filling, and examine them for differences when grown under organic versus conventional conditions.

Consumers prefer rice that isn’t broken. Therefore, head rice yield is an extremely important characteristic for rice farmers because millers are willing to pay more for rice kernels that are whole and not broken. Some cultivars consistently have greater head rice yield than others across different years and environmental conditions [3]. The two studies that examined head rice yield in this review found conflicting results for one cultivar each, grown in an organic versus the conventionally managed field. Thus, a conclusion on the effects of organic growing conditions on head rice yield can’t be made.

People in different global regions prefer cooked rice with a particular suite of textural properties. Rice with the preferred texture demands a premium price. Thus, it is important to understand the effects of growing conditions on rice cooked rice texture. There are fourteen aspects of cooked rice texture that are evaluated using trained sensory panels as well as various instrumental methods that are predictive of some of these characteristics [3]. One study in this review reported that three aspects of texture (i.e., slickness, hardness, and chewiness) were found by a trained sensory panel to be slightly influenced by organic versus conventional cultural management [19]. These differences were associated with grain protein content, which varies along with the dose of nitrogen applied during rice growth [19,46]. Thus, if differences are seen due to cultural management practices, they will likely be small and not be caused by the practices per se, but rather by changes in protein content caused by the difference in nitrogen application rates.

The primary predictor of cooked rice texture is the amount of amylose the grains contain and to a lesser degree the protein and lipid fractions (Fitzgerald 2004) [46]. Rice grains are classified according to amylose content: waxy (0%); very low (3–9%); low (10–19%); intermediate (20–25%); or high (>25%) [3]. The greater the amount of amylose the firmer and the less sticky the cooked rice is. Cultivars in the same amylose category are expected to have similar textural properties. The studies examined in this review indicate that organic production practices either don’t influence the quantity of amlyose in milled rice grain or have very little impact. When differences did occur, they were not large enough to move a cultivar from one amylose classification to another.

The food processing industry tests the viscous properties of rice using cooking stirring viscometers, typically a Rapid Visco Analyser (RVA) [47]. Some of these properties are related to aspects of cooked rice texture [48]. For example, the setback parameter is frequently used to predict cooked rice firmness/stickiness and pasting temperature is used when the rice will be included as a source of carbohydrates in brewing. The studies reviewed in this paper indicate that rice pasting properties will not likely be influenced by organic versus conventional farming practices for most cultivars. If differences are seen, they will probably be small and associated with grain protein content, which varies along with the dose of nitrogen applied during rice growth, as discussed above [46,48].

The aroma of ‘fragrant rice’ is an aspect of end-use quality of particular importance, as certain ethnic groups prefer it over nonfragrant rice; this impacts the market price of fragrant rice. For example, many different varieties of the jasmine style of rice are consumed in South East Asia, and many basmati styles of rice are consumed in South and Central Asia. Fragrant rice is commonly reported to smell similar to popcorn or bread-like due to a compound it contains called 2-acetyl-1-pyrroline and likely due to other compounds yet to be identified [49]. Various effects, such as genetics, environment, and cultural management practices, are reported to impact the level of 2-acetyl-1-pyrroline in fragrant rice [50,51,52]. Thus, it was surprising that Jasmine 85, a U.S. fragrant cultivar, had a similar aroma when grown using different cultural management practices, specifically organic and conventional [19]. Perhaps the difference lies in that all of the studies mentioned above were performed by measuring 2-acetyl-1-pyrroline using laboratory instrumentation, except for the one that evaluated Jasmine 85. The latter used a trained sensory panel to evaluate the “popcorn” aroma and several other aspects of rice aroma. The differences identified via instrumentation may be too small for humans to sense, and humans may also be smelling aromatic compounds other than 2-acetyl-1-pyrroline in the fragrant rice.

### 4.2. Healthfulness

Micronutrients, vitamins, and minerals, are essential for human health, and their deficiency in the diet remains a widespread problem, especially in low- and middle-income countries (Bailey et al., 2015) [53]. The articles examined in this review evaluated three minerals of particular importance for human health globally; Ca, Zn, and Fe. The latter two are minerals of particular importance to this review since reliance on milled rice with minimal dietary diversity contributes to Zn and Fe deficiency in developing countries [54]. Considerable variation in Zn and Fe levels exists within rice germplasm that is genotype dependent [55]. Iron levels vary from 6.9 to 22.3 mg/kg and zinc concentration ranges from 14.5 to 35.3 mg/kg in unpolished, brown rice. No trend in the effect of organic versus conventional management practices on Zn and Fe levels was found in the studies examined in this review. However, the data indicates that if organic production methods impact Zn and Fe levels of rice, the effects are small and less than the variation that exists between cultivars.

Interest in the effects of cultural management practices on gamma-oryzanol and tocols is of interest because these compounds have been proposed to have human health-beneficial properties [4,56]. Previous work suggested that in general the environment rice is grown in has a greater effect on gamma-oryzanol and tocol levels than genotype ([57]. Similarly, the studies examined in this review found that gamma-oryzanol and tocol levels were influenced by the environment and cultivar. No trend was found for the effects of organic versus conventional management methods. Although the data is limited, when there are effects of cultural management on gamma-oryzanol and tocols they will likely be small and less than the variation caused by differences in environment and cultivar.

### 4.3. Safety

Soil contamination with heavy metals has increased in certain regions of the world because of the following anthropogenic activities: urbanization, industrialization, mining, transportation, and agriculture [58,59]. Reports from several countries indicate that heavy metal concentrations (i.e., As, Cd, Cr, Hg, Ni, and Pb) in rice often exceed guidance values [60,61,62]. This elevated exposure to these metals creates an elevated risk to humans that rely on rice for a significant portion of their food. Consequently, there is an increasing need to mitigate the phytoaccumulation of heavy metals in rice.

The repeated use of inorganic fertilizers and metallo-pesticides is associated with increased levels of heavy metals in rice-growing soil [63]. Thus, it has been proposed that using organic cultural management practices may result in lower levels of heavy metals in soils. However, soils that are already contaminated with heavy metals may still pose a risk of contaminating rice with heavy metals even when under organic management. Increasing soil organic matter is one of the primary cultural management goals of organic farming, as it provides benefits such as an increase in the biodiversity of soil microflora, which in turn helps the soil retain nutrients. However, a potential drawback of an enhanced amount of organic matter in soil was reported by Zeng et al.) [64]. They reported that the bioavailability of Pb was positively correlated with the level of organic matter in the soil used to grow rice because of the effects of organic matter on element mobilization and bioavailability in soil.

In this review, none of the studies evaluated the same cultivars grown in the same environment. Rice purchased from grocery stores can be a blend of several cultivars that have been grown in fields with different levels of heavy metals, thus there is no way to fairly evaluate the effects of cultural management practices. In addition, these studies didn’t use statistical methods to evaluate the levels of heavy metals in the rice samples. Therefore, no conclusions about the effects of organic versus conventional cultural management methods on rice heavy metal content could be drawn.

Since 1960, the average yield of rice globally has more than doubled, as pesticides have increased by 15 to 20-fold [65]. Significant evidence exists that the use and especially the overuse of pesticides is associated with adverse effects on human health and non-target organisms such as birds, bees, and fish [66,67,68]. We conclude that the studies evaluated in this review found that organically produced rice grain was less likely to contain residues of the pesticides examined in the study than the rice grown using conventional methods. However, a problem with the design of one of the studies was that the reader wasn’t informed of how long the field studied had been under organic management. Over time some pesticides degrade and become nontoxic or less so [69]. The degradation rates of pesticides vary along with the soil microbial composition and other soil factors, such as pH and temperature [70]. Therefore, the length of time a field has been under organic management needs to be recorded in studies designed to evaluate the effect of this cultural management technique in comparison to conventional methods.

Mycotoxins are naturally occurring toxic contaminants found in cereal grains and other foods [71]. These secondary metabolites are made by fungi which, when consumed, have acute and long-term health risks for humans [72]. These fungi reportedly grow in rice when certain conditions occur during “particular crop seasons, cultivation regions, and agricultural practices (pre-harvest: paddy variety, crop residue management, and fertilizer application; post-harvest: means of transportation and delayed drying time)” [73,74]. A multi-year survey reported that the mycotoxin-producing fungi, *Fusarium proliferatum,* and *Aspergillus flavus*, were found more commonly in fields managed using a combination of fertilizers (organic and inorganic) or with crop debris compared to those fields that had only inorganic fertilizer applied [73]. This review found that most studies related to fungi and mycotoxin levels in rice-based food products and milled rice were difficult to interpret because the data wasn’t analyzed statistically. However, Juan et al. found ochratoxin A in some of the organically grown rice and rice products, but not in any of the products or rice from conventionally managed fields [32]. Also, the levels of nivalenol were significantly higher (*p* < 0.05) in organically grown rice (unmilled and milled) compared to the corresponding conventionally grown rice [37]. Although the evidence is limited, the literature suggests that rice and rice-based products may contain some fungi or some mycotoxins in particular global regions while their conventionally produced rice counterparts don’t.

## 5. Conclusions

This review provides tentative conclusions that food processing companies and consumers will not likely notice any aroma or processing quality differences between the same rice cultivars grown using organic farming practices compared to conventional methods. However, slight differences in cooked rice texture may be sensed due to differences in kernel protein content which is known to impact rice texture. Differences in rice grain protein content occur from exposure to different amounts of nitrogen, not likely due to organic sources of nitrogen versus conventional sources. There was insufficient evidence to evaluate the effect of organic production methods on chalk or milling yield. We conclude that the studies evaluated in this review found that organically produced rice grain was less likely to contain residues of the pesticides examined in the study than the rice grown using conventional methods. There was some evidence that organically grown rice is more likely to be contaminated with mycotoxin-producing fungi and some mycotoxins. Common shortcomings of some of the studies evaluated in this review were that they were poorly designed, with limited to no details of the cultural management practices used to grow the rice studied, cultivars were not named, and the data wasn’t analyzed statistically. Future related research should use fields that have been under organic management for more than two years, take place in more than one environment, and have a variety of soil types.

## Figures and Tables

**Figure 1 foods-12-00073-f001:**
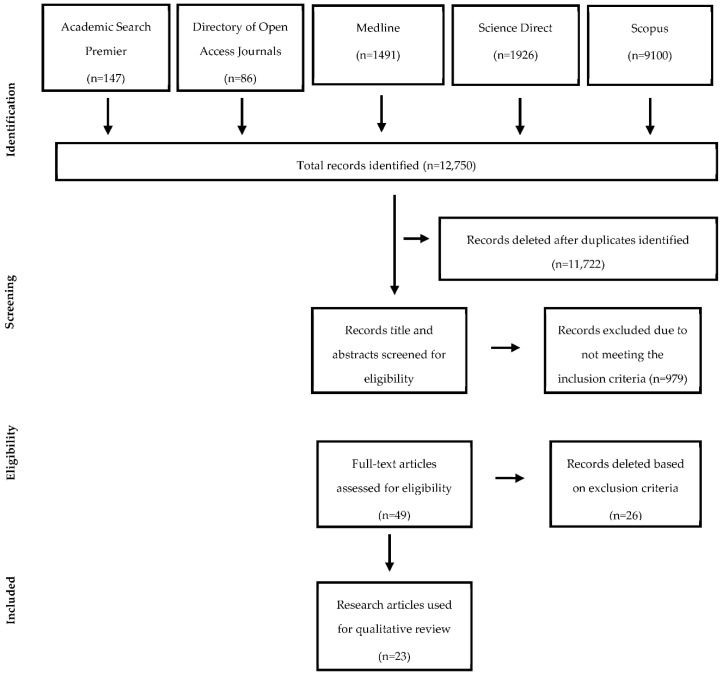
Flow chart created from a systematic review of organic versus conventional rice grain using the preferred reporting items for systematic reviews (PRISMA) methodology [16].

**Table 1 foods-12-00073-t001:** Full-text articles deleted from the total records based on exclusion criteria.

Exclusion Criteria	# of Articles Identified
Full paper not written in English	3
Not specifically related to the topic	17
No conventional rice used for comparison to organic	2
Not a research article	2
Duplicate articles	1
Methodology unclear	1
TOTAL	26

**Table 2 foods-12-00073-t002:** Organic versus conventional rice grain-related studies were identified using the preferred reporting items for systematic reviews (PRISMA) methodology.

COUNTRY ^1^	# of STUDIES	CATEGORIES OF TRAITS EXAMINED ^2^		
		End-Use Quality	Healthfulness	Safety
Afghanistan	1	1		
Brazil	5	1	3	5
India	2	1		1
Italy	1			1
Malaysia	1	1		
Philippines	1	1	1	1
South Korea	2		1	1
Spain & Portugal	3			3
Thailand	4	2	3	1
United States	3	1	1	1
TOTAL	23	8	9	14

**^1^** Countries where the rice samples used in the studies were grown or purchased. **^2^** Some studies evaluated traits in more than one category.

**Table 3 foods-12-00073-t003:** Organic versus conventional rice grain-related studies identified using the PRISMA methodology: materials, methods, and findings summary.

Materials and Study Design	Traits	Organic Rice Production Methods	Conventional Rice Production Methods	Main Findings	Citation
“IRGA 410” **^1^** grown in Rio Grande Do Sul, Brazil.	Brown rice proximate analysis, phenolics, amylose content, cooking time, head rice yield, and stained grains during storage. Mycotoxin analysis. Fungal incidence in rough rice.	Seeding rate 90 kg/h. Water management same as conventional No other inputs were reported.	Seeding rate 90 kg/h. Water management same as organic. Urea was applied at 140 kg/ha in dry soil, before the appearance of the 3rd leaf, and at 60 kg/ha at the beginning of panicle development. Two fungicide applications of PrimoR©300 mL/ha. One application of TalismanR© 250 mL/ha.	Conventional unmilled rice had greater protein, lipid, and ash content, and higher head rice yield. It also had greater Aspergillus sp. after storage. Organic thrashed rice had greater total carbohydrates, soluble protein, amylose content, free phenolics, and phenolic acids. It also had a greater percentage of stained grains and Bipolaris sp. after storage.	[23]
Organic (n = 17) and conventional (n = 33) rice samples were purchased from different Brazilian producers (50 different brands) in different cities.	Cu, Zn, Mg, B, P, Mo, As, Pb, Cd, Mn, Se, Co, Cr, Ba, Rb, Fe, Ca, La, and Ce contents.	Certified organic by the Brazilian IBD-Agricultural and Food Inspections and Certifications which is accredited by the International Federation of Organic Agriculture Movements.	No description of conventional methods was reported.	Ca levels are significantly higher in all org. samples but one. Cd is higher in conventional samples. No difference in As levels. Statistical differences in other minerals weren’t analyzed.	[27]
Rice samples purchased from markets in Brazil. 12 milled rice, 10 parboiled milled rice samples, 2 unmilled, and 5 unmilled parboiled samples. One organic milled and one parboiled organic sample.	As	No organic production methods were described.	No conventional production methods were described.	The mean of total As for the milled samples was greater than in the organic milled sample (222.9 and 161.6 ng g^−1^, respectively). The mean of total As for parboiled white samples was greater than the parboiled organic sample (214.9 and 174.1 ng g^−1^, respectively).	[28]
Randomized block design, 4 replications during 3 years. “Cypress”, “Bengal”, “Jasmine 85”, “Jacinto”, and “Neches” grown in adjacent fields that had been fallow for 2 years.	Paste viscosity properties; amylose, protein, Ca, Fe, P, K, Mg, Mn, and Zn contents. Volatile compounds and sensory evaluation.	Chicken litter was applied (76 kg/ha N, 25 kg/ha P, and 25 kg/ha K), a microbial product of trace minerals (67 kg/ha), and a microbial soil activator (33.8 L/ha) was applied prior to planting. Seed treated with hurnic acid (5 mLJkg of seed), a microbial inoculant (mL/100 g of seed), and manganese sulfate (20 g/kg of seed). Before the flood, a side dress of chicken litter was applied (126 kg/ha N, 42 kg/ha P, and 42 kg/ha K). At panicle differentiation, a fish emulsion was applied as a foliar spray (16.5 L/ha) for insect control.	Urea nitrogen (56 kg/ha, 90 kg/ha, and 78 kg/ha) was applied at planting, flood, and panicle differentiation, for the 100% nitrogen treatment. The 50% nitrogen treatment was applied using half of the rate of urea (112 kg of N/ha). Standard chemical management practices were used to control weeds and insects in conventionally managed plots.	Milled kernel protein content in conventional 100% urea nitrogen samples > other treatments. Little to no difference in amylose and mineral contents between treatments. Differences in pasting properties were found associated with kernel protein content. No differences in flavor attributes were found via the sensory panel or in volatile compounds. No differences in microbially produced volatile compounds were found.	[19]
Randomized block design with two field replications and two years. “Cocodrie’, ‘Presidio’ “Sierra”, “Giant Embryo” (GSOR 25), “IL 121-1-1” and “Sigoendab”	Total phenolics, flavonoids, tocols, and γ-oryzanol	Certified organic fields followed two years of fallow and a clover/ryegrass winter cover crop. Nature Safe 13-0-0 fertilizer (1681 kg/ha) was applied just prior to planting in both years. Year one seed was drilled with two seeding rates, 112 and 135 kg/ha. In year two, water-seeding was done with 202 kg/ha.	Following two years of fallow, seeds were drilled at 112 kg/ha. A total of 224 kg/ha of nitrogen as urea (46-0-0) was applied with a three-way split: at planting (56 kg/ha), at permanent flood (90 kg/ha), and at panicle differentiation (78 kg/ha).	The growing environment affected the concentrations of most traits, especially the tocols and γ-oryzanol. The effect of conventional versus organic management systems had the lowest effect on the phytochemical levels of the year, replication, and management system.	[18]
“Dongjin” was obtained from one organic field and one conventional field. A sampling of organic rice was done in the central region of the block.	Total gamma-oryzanol compounds.	The field had been organically managed for 5 years. Independent groundwater was used. The green manure crop, Vicia villosa Roty), was used in the organic plot.	Field managed conventionally for 10 years. Fertilization included (N/P/K = 9:4.5:5.7, *w*/*w*, 202 kg/ha). Weed prevention performed using herbicide [1.0% fentrazamide and 0.07% pyrazosulfuron-ethyl, suspension concentrate, 321 green area index (gai)/ha] Pesticide (2% chlorantraniliprole, 16 gai/ha)	Organic brown rice had higher total gamma-oryzanol than conventional (65.6 ± 2.7 mg/100 g and 60.2 ± 1.8 mg/100 g respectively)	[20]
Rice-based foods from Italian stores: flours, biscuits, and rice flakes (13 conventional and 11 organic products).	Deoxynivalenol, fumonism B1, and fumonism B2	No organic production methods were described.	No conventional production methods were described.	Deoxynivalenol was identified in more organic rice foods than conventional (91% vs. 85%). Furmonism B1 occurred in 36% of organic rice foods versus 23% of conventional. Furmonism B2 occurred in 38% of conventional vs. 45% of organic rice food products.	[29]
Rice samples and rice-based foods were collected from cultivars and markets in Spain. 64 were conventional and 20 were organic.	Ochratoxin A	No organic production methods were described.	No conventional production methods were described.	Ochratoxin A was found in a greater % of organic rice and products compared to those grown using conventional methods.	[30]
10 baby cereals are made using conventional rice and 3 produced using organic rice obtained from manufacturers.	Hg and total As.	No organic production methods were described.	No conventional production methods were described.	Hg was higher in organic rice cereal compared to conventional rice cereal (4.54 and 4.39 μg/kg, respectively). As was higher in organic rice cereal compared to conventional rice cereal (154.9 and 96.3 μg/kg, respectively)	[31]
9 organic and 12 conventionally produced rice samples and rice foods bought from markets in Spain and Portugal.	Ochratoxin A	No organic production methods were described.	No conventional production methods were described.	Ochratoxin A found in 4/9 organic samples and 0/12 conventionally produced rice foods.	[32]
10 infant rice cereals made with commercially produced milled rice and 10 infant rice cereals made with organically produced unmilled rice.	Total As and inorganic As (i-As).	No organic production methods were described.	No conventional production methods were described.	All samples had identifiable As and iAs. No significant difference between organic and conventionally produced rice cereals.	[33]
Randomized block design, one year “Attai-1”, five cultural management treatments, four replications.	Unmilled kernels: perfect grains, broken grains and amylose, protein, and lipid contents.	The recommended dose for traditional farming (120 kg/ha urea and 100 kg/ha diammonium hydrogen phosphate)	Animal manure (5 tons/ha) (AM), animal manure + 50% recommended dose of nitrogen and phosphorus (AMRD), sawdust + green leaves (5 tons per ha) (SD), sawdust + leaves and 50% recommended dose of nitrogen and phosphorus (SDRD).	Greater whole kernels in AMRD compared to other treatments. No significant difference in broken grains. Amylose content in AMRD and SDRD > AM and SD > RD. Protein content in AMRD and AM and SDRD > RD and SD. Lipid content in AMRD was > than in the other treatments.	[25]
Unmilled rice is grown organically and conventionally in Thailand.	Kernel: elongation ratio, hardness, and color, water uptake capacity, and starch crystallinity	No organic production methods were described.	No conventional production methods were described.	Higher elongation ratio for conventional than organic (1.10 and 1.06, respectively, after six months of storage). Conventional rice was harder and darker compared to organic rice. Higher water uptake capacity in cooked organic rice than conventional after six months (4.52% and 4.47%, respectively). Crystallinity increased for both organic and conventionally produced rice during ageing	[34]
One organic and one conventional rice system is used to produce unmilled “Sungyod” rice.	Kernel: size, weight, and color. Free fatty acids, proximate analysis. amylose and anthocyanin content. Kernel swelling power and H_2_0 solubility of starch. Pasting gelatinization enthalpy and temperature. X-ray diffraction patterns.	No organic production methods were described.	No conventional production methods were described.	Conventional rice had greater grain length and breadth than organic (0.62 and 0.61 cm, respectively). Conventional rice weight was higher than organic rice (1.44 and 1.42 g/100 grains, respectively). No difference in kernel color, free fatty acid content, and crystallinity pattern. Conventional rice protein content > organic (7.02 and 5.64%, respectively). Conventional rice lipid content > organic rice (2.59 and 2.48%, respectively). Conventional rice amylose content (16.27 and 15.32%, respectively). Conventional rice anthocyanin content > organic rice (15.6 and 14.66 mg cyanidin-3-glucoside/100 g, respectively). Organic rice showed higher swelling power and H_2_0 solubility than conventional rice. Conventional rice had a higher setback value than conventional rice (129.93 and 123.65 RVU, respectively). Convention rice had a higher transition temperature and gelatinization enthalpy than organic rice.	[24]
One unmilled organic Jasmine rice sample and one unmilled milled conventional Jasmine rice sample.	Serum cholesterol, triglycerides, HDL-C, and LDL-C levels in rats after a feeding trial.	No organic production methods were described.	No conventional production methods were described.	No significant difference in rat lipids between those fed conventional versus organic rice.	[35]
One organic unmilled rice sample and one conventional unmilled rice sample were supplied by a farming cooperative in Thailand.	Rat protein efficiency (PER) level. Pesticide residues (carbofuran, methyl parathion, p-nitrophenol, and beta-cyfluthrin) in rice and rat serum after a 28-day feeding trial.	No organic production methods were described.	No conventional production methods were described.	Carbofuran, methyl parathion, and B- cyfluthrin were not present in any rat serum samples or in rice samples. P-nitrophenol was found in both samples but not in rat blood serum. Data wasn’t analyzed to determine if levels were different between the conventional and the organic sample. No significant effect of organic rice on PER compared to conventional rice was found.	[36]
39 milled conventional rice samples and 37 milled organic rice. 26 conventional unmilled rice and 22 organic unmilled samples. All samples were obtained from stores in Korea.	Five mycotoxins: 8-ketotrichothecenes (deoxynivalenol (DON), nivalenol (NIV), 3-acetyldeoxynivalenol (3ADON), 15-acetyldeoxynivalenol (15ADON) and fusarenone-X (FUS-X)	No organic production methods were described.	No conventional production methods were described.	Contamination of NIV was greater in organic samples compared to their conventional counterparts. DON was detected in 19% and 41% of organic milled and unmilled rice, respectively, and 10% and 27% in conventional milled and brown samples. 3ADON, 15 ADON, and FUS-X were low in all samples, and no difference was found between conventional and organic samples.	[37]
Samples purchased rice at stores in Brazil. Not enriched. Rice wasn’t enriched. 5 conventional milled rice and 2 organic milled rice samples. 2 conventional unmilled and 3 organic unmilled samples.	As, Cd, Pb, Ti, Sb, Co, Cu, Mn, Se, Zn, Cr, Ni, and Mo.	No organic production methods were described.	No conventional production methods were described.	Hg, Sb, and Tl were not detected in any samples. Cr was highest in the milled conventional rice (641 µg kg^−1^). Conventional milled rice mean were as follows: (As, Cd, Ni, Pb, Zn, Mn, Cu, Se, Co, Mn, and Mo µg/k (164, 18.9, 130, 57.4, 17.9, 14.4, 1.79, 66.9, 25.6, 14.4, and 511, respectively). Conventional unmilled rice means were as follows: (As, Cd, Ni, Pb, Zn, Mn, Cu, Se, Co, Mn, and Mo (293,16.8,140, 109, 23.5, 31.4, 2.34, 84.7, 36.2,31 and 4,344 µg/k, respectively). Organic unmilled rice means were as follows: (As, Cd, Ni, Pb, Zn, Mn, Cu, Se, Co, Mn, and Mo (215, 13.4 179, 119, 23.6, 29.8, 2.17, 107, 44.1, and 29.8,367 µg/kg, respectively). Organic milled rice means were as follows: As, Cd, Ni, Pb, Zn, Mn, Cu, Se, Co, Mn, and Mo (149, 19.6, 98.9, 39.7, 15.7, 8.2, 1.43, 57.1, 11.5, 8.2, and 361 µg/kg, respectively). Organic and conventional milled samples did not differ in the amount of any elements measured. Organic and conventional unmilled samples did not differ in the amount of any elements measured.	[38]
Rice samples were collected from 10 organic and 10 conventional farms from all 16 agro-climatic zones in India.	Four groups of pesticides: organochlorine, carbamates, organophosphorous, and pyrethrites.	Organic farms are certified by each local state government.v No description of production methods was provided.	Conventional farms were adjacent to organic farms. No description of production methods was provided.	Carbamates and pyrithroid were found in conventional rice samples. No traces of pesticides were found the in the organic rice samples.	[39]
Rice samples from grocery stores in Brazil. Organic milled rice (n = 18), conventional milled rice (n = 11), organic husked rice (n = 12), conventional husked rice (n = 15), and specialty types (n = 13).	Organic and inorganic As.	No organic production methods were described.	No conventional production methods were described.	No difference in total As between conventional husked and organic husked samples. No difference in total As between conventional milled and organic milled samples. Inorganic As is 45% greater in organic milled rice compared to conventional milled rice and 41% greater in organic husked versus conventional husked rice.	[40]
Rice was purchased from a supermarket in Malaysia. “Bario” is grown organically. Basmati rice is grown conventionally in Pakistan. Noodles are stored for 3 days.	Rice noodle color, tensile strength, elasticity, cooking loss, rehydration ratio, aerobic plate content (APC), and total yeast and mold contents (TYMC). Water activity and sensory evaluation.	No organic production methods were described.	No conventional production methods were described.	Both rice noodles became darker in color during storage. Bario noodles had higher tensile strength due to higher amylose content compared to Basmati noodles (46.33 and 36.33 kPa, respectively, on day 0). Bario noodle was higher in elasticity than basmati (13.19 and 7.89 kPa, respectively). Basmati rice noodles had a higher cooking loss compared to Bario ( 7.14% vs. 3.89% respectively). Bario rice noodles had a higher rehydration ratio than basmati rice noodles (3.89 and 3.71). Higher APC in Basmati rice than in Bario, but both were acceptable after three days of storage. TYMC was higher in Basmati than in Bario, but both were above recommended amount after day two. Water activity was higher in Basmati rice than in Bario rice (0.82–0.87 and 0.80–0.83, respectively, after three days). Bario rice was most accepted and concluded to have better quality than Basmati rice.	[26]
The study was conducted for two seasons at PhilRice Philipines. A splitplot experimental design was used: two main plots and three subplots with four replications. “NSIC Rc146” was planted in the dry season (1st crop of organic farming) and NSIC Rc160 in the wet season (2nd crop of organic farming).	Tocols, gamma-oryzanol, and total phenolics. Head rice yield, kernel length, breadth, and ratio. Amylose and protein content. Kernel color.	The two main plots consisted of “with pesticide” and “without pesticide” treatments. The subplots included control, organic fertilizer, and inorganic fertilizer with a quadruplicate plot size of 10 × 4 m per plot. The organic fertilizer used was compost at 3 tons/ha (13-2-17-16 S) applied 3 d before transplanting. The NSIC Rc160 crop was applied with Bayluscide EC 250 (250 g/L Niclosamide, 200 g/L methyl isobutylketone, 100 g/L isobutanol, 1 L/ha) molluscicide 1 DAT, and Brodan 3.51 EC insecticide at 39 and 83 DAT.	The two main plots consisted of “with pesticide” and “without pesticide” treatments. The subplots included control, organic fertilizer, and inorganic fertilizer with a quadruplicate plot size of 10 × 4 m per plot. Inorganic fertilizer was applied 21-0-0-24 S at 13 DAT, urea (45-0-0) at 28 DAT, 34-0-0 at 41 DAT, and 20-0-0 at 51 DAT for a total of 120-0-0-24 S. The NSIC Rc146 crop was treated with Furadan (3 g/kg Carbofuran, 16.7–33.3 kg/ha) 28 d after transplanting (DAT) and with Brodan 3.51 EC (210 g/L Chlorpyrifos + 105 g/L BPMC (Fenobucarb), 2.5–3.5 tablespoons (45 ± 7.5 mL) 16 L −1, 120 mL/ha) insecticide 69 DAT.	Pesticide application had no effect on tocols and gamma-oryzanol levels. Organic milled rice had lower total and γ-oryzanol than conventional milled rice with applied pesticides. NSIC Rc160, organic brown rice with pesticide had higher contents of total tocols than inorganic unmilled rice with pesticide. Organic milled rice had lower total and gamma-oryzanol compared to conventional rice. Organic fertilizer resulted in lower total phenolics. No difference in milling quality, grain color, apparent amylose content, and alkali spreading value between organic and conventional rice was found. Organic rice was lower in protein content compared to conventionally grown rice.	[21]
Randomized block design with three replications of “Pusa Basmati-1” Rice.	Head rice recovery (HRR). Kernel length (KL), kernel breadth (KB), and the ratio between the two (KL:KB) before and after cooking.	The soil of the experimental field was a sandy clay loam, having 52.8% sand, 21.5% silt, and 25.7% clay. It contained 0.56% organic C, 163.2 kg/ha 71 NaOH-KMnO_4_ hydrolysable N, 15.5 kg/ha 71 0.5 N NaHCO_3_, extractable P, and 232.4 kg/ha 71 N NH_4_AOC extractable K and had a pH value of 8.2. Seven combinations of organic sources (Farm yard manure (FYM), Sesbania green manuring (SGM), FYM + blue-green algae (BGA), SGM + BGA, FYM + SGM, FYM + SGM + BGA and FYM + SGM + BGA + PSB). FYM was applied at 10 tons/h at the time of final puddling. Sesbania was grown for 60 days and incorporated 5 days before transplanting. BGA was inoculated 10 days after transplanting of rice, whereas PSB was inoculated by dipping the roots of rice seedlings in the slurry of Pseudomonas striata culture.	The soil of the experimental field was a sandy clay loam, having 52.8% sand, 21.5% silt, and 25.7% clay. It contained 0.56% organic C, 163.2 kg/ha 71 NaOH-KMnO_4_ hydrolysable N, 15.5 kg/ha 71 0.5 N NaHCO_3_, extractable P, and 232.4 kg/ha 71 N NH_4_AOC extractable K and had a pH value of 8.2. Four rates of inorganic fertilizers (control, 60 kg N+13 kg P + 17 kg K/ha, 120 kg N + 26 kg P + 34 kg K/ha, and 180 kg N + 39 kg P + 51 kg K/ha).	The different conventional management treatments did not affect HRR, KL, KB, and KL/KB ratio before or after cooking. Organic manure increased HRR in comparison to conventional treatments. KL hasn’t affected the organic treatments. KB in the organic treatments was greater than in the conventional treatments. The data wasn’t analyzed statistically.	[22]

**^1^** Names in quotation marks denote that these are the names of rice cultivars.

## Data Availability

Not applicable.

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
