# Peer review of "Organic Rice Production Practices: Effects on Grain End-Use Quality, Healthfulness, and Safety"

_foods, 2022, doi:10.3390/foods12010073_

Round 1
Reviewer 1 Report
Dear Editor,
The manuscript is well-written and scientifically novel and sound. I congratulate the authors for carrying an excellent outcome from this research.
Author Response
Dear Reviewer,
Thank you very much for your thoughtful review and comments.
Reviewer 2 Report
1. In the Fig.1 some texts are missing. Check it.
2. In place of writing “Only Chen and McClung, Champagne et al., Cho et al., Tuano et al., and VanQuyen et al., reported in some detail the agronomic practices used to produce the rice for their 192 study [18-22].” write “Only Chen and McClung [18], Champagne et al.[19], Cho et al. [20], Tuano et al. [21], and Van Quyen et al. [22], reported in some detail the agronomic practices used to produce the rice for their study”.
3. There is lot of issue related to punctuation such as in the statement “Seeding rate 90 kg/h. Water management same as conventional No other inputs reported.” There must be ‘.’ At the end conventional. Check it throughout the manuscript.
4. There is big issue related to citing the references in the text. Somewhere authors written as “Juan et al., 2008” while at other place they used the format “Champagne et al. 2007”.
5. Either use “Alves et al. 2017” or “Alves et al. [19]” pattern to cite the references in the text. Not both.
6. Provide the future extension with proper references.
7. Page number is missing in some references. In the reference “Shimbo, S.; Zhang, Z.; Watanabe, T.; Nakatsuka, H.; Matsuda-Inoguchi, N.; Higashikawa, K.; Ikeda, M. Cadmium and lead 853 contents in rice and other cereal products in Japan in 1998–2000. Science of The Total Environment 2001, 281, 165, DOI 854 10.1016/s0048-9697(01)00844-0.” Page number is 165-175. Cross check all the references.
8. DOI is given to the few references only. Check it.
9. Follow the format of journal.
Author Response
Dear Reviewers,
Thank you very much for your thoughtful review and comments. Our response to your comments and the changes made are addressed below.
- In the Fig.1 some texts are missing. Check it.
The top three text boxes were made bigger so the n= information can now be read.
- In place of writing “Only Chen and McClung, Champagne et al., Cho et al., Tuano et al., and VanQuyen et al., reported in some detail the agronomic practices used to produce the rice for their 192 study [18-22].” write “Only Chen and McClung [18], Champagne et al.[19], Cho et al. [20], Tuano et al. [21], and Van Quyen et al. [22], reported in some detail the agronomic practices used to produce the rice for their study”.
We corrected this in the text.
- There is lot of issue related to punctuation such as in the statement “Seeding rate 90 kg/h. Water management same as conventional No other inputs reported.” There must be ‘.’ At the end conventional. Check it throughout the manuscript.
So sorry, we don’t understand this comment. “Seeding rate 90 kg/h. Water management same as conventional No other inputs reported” is within Table 3, and thus, the verbiage must be short statements rather than complete sentences.
- There is big issue related to citing the references in the text. Somewhere authors written as “Juan et al., 2008” while at other place they used the format “Champagne et al. 2007”.
We corrected this in the text.
- Either use “Alves et al. 2017” or “Alves et al. [19]” pattern to cite the references in the text. Not both.
We corrected this in the text.
- Provide the future extension with proper references.
So sorry, we don’t understand this comment.
- Page number is missing in some references. In the reference “Shimbo, S.; Zhang, Z.; Watanabe, T.; Nakatsuka, H.; Matsuda-Inoguchi, N.; Higashikawa, K.; Ikeda, M. Cadmium and lead 853 contents in rice and other cereal products in Japan in 1998–2000. Science of The Total Environment 2001, 281, 165, DOI 854 10.1016/s0048-9697(01)00844-0.” Page number is 165-175. Cross check all the references.
- DOI is given to the few references only. Check it.
We corrected this in the text.
- Follow the format of journal.
We corrected this in the text.
We had Track Changes on, but it started to greatly slow down our ability to make changes to the references. Thus, after reference 25 we turned it off, but we did make changes to the remainder of the references.
Reviewer 3 Report
This is an interesting manuscript. There are some recommendations, as follows:
1. Why were only the previous 25 years of research chosen as inclusion criteria?
2. There is a typo on line 126.
3. Figure 1 needs to be improved.
4. Providing a table in which to list the 23 selected references for systematic review may improve the readability of the manuscript.
Author Response
Dear Reviewer,
Thank you very much for your thoughtful review and comments. Our response to your comments and the changes made are addressed below.
- Why were only the previous 25 years of research chosen as inclusion criteria?
The following was added to the 2.1. Inclusion Criteria section.
This time period was selected because it was in 2002 that the regulations under the U.S. Organic Foods Production Act were implemented, and other countries, such as Brazil, adopted similar regulations sometime after this [15].
- There is a typo on line 126.
The extra period was removed.
- Figure 1 needs to be improved.
The top three text boxes were made bigger so the n= information can now be read.
- Providing a table in which to list the 23 selected references for systematic review may improve the readability of the manuscript.
Table 3 contains this information.